# *N,Nʹ*-Diarylurea Derivatives (CTPPU) Inhibited NSCLC Cell Growth and Induced Cell Cycle Arrest through Akt/GSK-3β/c-Myc Signaling Pathway

**DOI:** 10.3390/ijms24021357

**Published:** 2023-01-10

**Authors:** Sunisa Thongsom, Satapat Racha, Zin Zin Ei, Korrakod Petsri, Nithikoon Aksorn, Supakarn Chamni, Vitsarut Panpuang, Hongbin Zou, Pithi Chanvorachote

**Affiliations:** 1Center of Excellence in Cancer Cell and Molecular Biology, Faculty of Pharmaceutical Sciences, Chulalongkorn University, Bangkok 10330, Thailand; 2Department of Pharmacology and Physiology, Faculty of Pharmaceutical Sciences, Chulalongkorn University, Bangkok 10330, Thailand; 3Interdisciplinary Program in Pharmacology, Graduate School, Chulalongkorn University, Bangkok 10330, Thailand; 4Department of Clinical Pathology, Faculty of Medicine Vajira Hospital, Navamindradhiraj University, Bangkok 10300, Thailand; 5Department of Pharmacognosy and Pharmaceutical Botany, Chulalongkorn University, Bangkok 10330, Thailand; 6Natural Products and Nanoparticles Research Unit (NP2), Faculty of Pharmaceutical Sciences, Chulalongkorn University, Bangkok 10330, Thailand; 7College of Pharmaceutical Sciences, Zhejiang University, Hangzhou 310058, China

**Keywords:** *N,Nʹ*-diarylureas, NSCLC, Akt, cell cycle progression, c-Myc

## Abstract

Lung cancer is one of the most common malignancies worldwide. Non-small-cell lung cancer (NSCLC) accounts for more than 80% of lung cancers, shows chemotherapy resistance, metastasis, and relapse. The phosphatidylinositol-3 kinase (PI3K)/Akt pathway has been implicated in the carcinogenesis and disease progression of NSCLC, suggesting that it may be a promising therapeutic target for cancer therapy. Although phenylurea derivatives have been reported as potent multiple kinase inhibitors, novel unsymmetrical *N,Nʹ*-diarylurea derivatives targeting the PI3K/Akt pathway in NSCLC cells remain unknown. Methods: *N,Nʹ*-substituted phenylurea derivatives CTPPU and CT-(4-OH)-PU were investigated for their anticancer proliferative activity against three NSCLC cell lines (H460, A549, and H292) by 3-(4,5-dimethythiazol-2-yl)-2,5-diphenyl tetrazolium bromide, colony formation, Hoechst33342/PI staining assays, and apoptosis analysis. The protein expressions of Akt pathway-related proteins in response to CTPPU or CT-(4-OH)-PU were detected by Western blot analysis. The Kyoto Encyclopedia of Genes and Genomes mapper was used to identify the possible signaling pathways in NSCLC treated with CTPPU. The cell cycle was analyzed by flow cytometry. Molecular docking was used to investigate the possible binding interaction of CTPPU with Akt, the mammalian target of rapamycin complex 2 (mTORC2), and PI3Ks. Immunofluorescence and Western blot analysis were used to validate our prediction. Results: The cytotoxicity of CTPPU was two-fold higher than that of CT-(4-OH)-PU for all NSCLC cell lines. Similarly, the non-cytotoxic concentration of CTPPU (25 µM) dramatically inhibited the colony formation of NSCLC cells, whereas its relative analog CT-(4-OH)-PU had no effect. Protein analysis revealed that Akt and its downstream effectors, namely, phosphorylated glycogen synthase kinase (GSK)-3β (Ser9), β-catenin, and c-Myc, were reduced in response to CTPPU treatment, which suggested the targeting of Akt-dependent pathway, whereas CT-(4-OH)-PU had no effect on such cell growth regulatory signals. CTPPU induced G1/S cell cycle arrest in lung cancer cells. Immunofluorescence revealed that CTPPU decreased p-Akt and total Akt protein levels, which implied the effect of the compound on protein activity and stability. Next, we utilized in silico molecular docking analysis to reveal the potential molecular targets of CTPPU, and the results showed that the compound could specifically bind to the allosteric pocket of Akt and three sites of mTORC2 (catalytic site, A-site, and I-site), with a binding affinity greater than that of reference compounds. The compound cannot bind to PI3K, an upstream regulator of the Akt pathway. The effect of CTPPU on PI3K and Akt was confirmed. This finding indicated that the compound could decrease p-Akt but caused no effect on p-PI3K. Conclusions: The results indicate that CTPPU significantly inhibits NSCLC cell proliferation by inducing G1/S cell cycle arrest via the Akt/GSK-3β/c-Myc signaling pathway. Molecular docking revealed that CTPPU could interact with Akt and mTORC2 molecules with a high binding affinity. These data indicate that CTPPU is a potential novel alternative therapeutic approach for NSCLC.

## 1. Introduction

Lung cancer is the most common cause of cancer-related deaths and the second most commonly reported malignancy worldwide [1]. The two most commonly reported subtypes of lung cancer are non-small-cell lung cancer (NSCLC) and small-cell lung cancer, with NSCLC accounting for 80–85% [2]. The prognosis is generally poor, with 5-year survival rates remaining very low at 15%, and the majority of patients with NSCLC are diagnosed at the advanced stage, which results in the limited availability of effective treatments [1,3]. A number of studies attempted to discover strategies for cancer treatment, and given the advancement in treatment options, the death rate of cancer has declined in the past decade [4]. Nevertheless, conventional therapies are frequently found to be effective for certain malignancies only, and highly aggressive cancers, such as lung cancer, can resist chemotherapy and exhibit metastasis and relapse [5,6].

Currently, the lung cancer treatment paradigm has shifted to targeted therapy. Several studies demonstrated that targeted treatment could increase the overall survival, progression-free survival, and response rate of cancer patients [7]. In addition, compared to chemotherapy, targeted therapy is more selective since it spares non-target cells and has fewer adverse side effects [8]. The phosphatidylinositol-3-kinase (PI3K)/Akt axis plays a crucial part in cancer development and progression [9]. This pathway is frequently and constantly activated in NSCLC [10] and is associated with oncogenesis, drug resistance, and disease relapse [9,11,12]. Several drugs targeting the PI3K/Akt signaling pathway [13], including buparlisib (PI3K inhibitor), MK2206 (Akt inhibitor), and perifosine (dual PI3K/Akt inhibitor), are undergoing clinical trials as lung cancer treatments [14,15,16].

Akt is a significant downstream effector of PI3K [9]. Targeted compounds against Akt and its downstream mediators have garnered attention as they may benefit the clinical outcomes of the disease [17]. Multiple preclinical studies of natural compounds with potent inhibiting effects on this signaling pathway have been reported [18,19,20]. Zhang et al. reported that erianin, a small-molecule benzyl compound derived from *Dendrobium chrysotoxum Lindl*, exhibited anticancer activity in lung cancer cells due to the induction of apoptosis, G2/M phase arrest, and the inhibition of the PI3K/Akt/mammalian target of rapamycin (mTOR) pathway in vitro and in vivo [21]. In previous studies, we have reported that c-Myc–targeting compound *N,N-bis* (5-ethyl-2-hydroxybenzyl) methylamine can induce drug-resistance lung cancer cells to undergo apoptosis [22]. In addition, gigantol, a natural product-derived compound, inhibits PI3K/Akt signal and can deplete tumor growth and destabilize tumors in vivo [23]. These pieces of evidence make Akt and c-Myc promising targets for lung cancer therapy.

In recent years, unsymmetrical *N,Nʹ*-diarylureas have received substantial attention due to their growing application in drug design and medicinal chemistry, including anticonvulsant, antibacterial, antiviral, anti-inflammatory, and anticancer activities [24,25,26]. In addition, diarylurea derivatives have been an important topic in the research and design of anticancer drugs due to their potent multiple kinase inhibitors, such as rapidly accelerated fibrosarcoma kinases, vascular endothelial growth factor receptors, platelet-derived growth factor receptors, FMS-like tyrosine kinase-3, and receptor tyrosine kinases (RTKs) [25]. Over the past decade, several diarylurea derivatives, such as sorafenib, lenvatinib, linifanib, and tivozanib, have been approved as anticancer drugs and have passed or are currently in clinical trials for cancer treatment [27]. Regarding the molecular pharmacological research area, the discovery of the lead compound targeting proteins of interest, along with the knowledge of the structure-activity relationship (SAR), can facilitate the development of novel compounds for cancer therapy.

Herein, we utilized our efforts to carry out an SAR study aimed at studying the effects of *N,Nʹ*-substituted phenylurea derivatives on the Akt-associated pathway in NSCLC cells. With the chemical modification of critical moiety on the core structure of the compound and computational molecular docking analysis, we predicted the key part of the compound molecule interacting with the Akt protein, which may benefit its further development and its associate analogs.

## 2. Results

### 2.1. N,Nʹ-Substituted Phenylurea Derivative Compounds

We previously reported the synthesis of unsymmetrically *N,Nʹ*-substituted phenylurea derivatives [28]. In this study, two novel phenylurea derivatives with similar chemical structures, *N*-[4-chloro-3-(trifluoromethyl)phenyl]-*Nʹ*-phenylurea (CTPPU) and *N*-[4-chloro-3-(trifluoromethyl)phenyl]-*Nʹ*-(4-hydroxyphenyl)urea (CTP-(4-OH)-PU; Figure 1A) were synthesized to evaluate their anticancer effects on NSCLC cells. The synthesis of target compounds is illustrated in Figure 1B.

### 2.2. Cytotoxicity and Antiproliferative Effect of N,Nʹ-Substituted Phenylurea Derivatives on NSCLC Cells

To evaluate the anticancer activity of *N,Nʹ*-substituted phenylurea derivatives (CTPPU and CTP-(4-OH)-PU) on human NSCLC cells, we performed a cell viability test using 3-(4,5-dimethylthiazol-2-yl)-2,5-diphenyltetrazolium bromide (MTT) assay on three human NSCLC cell lines (H460, A549, and H292) and treated them with various concentrations of *N,Nʹ*-substituted phenylurea derivatives (0–100 µM) for 24 h. Compared with the untreated cells, treatment with CTPPU markedly inhibited NSCLC cell viability, with half maximal inhibitory concentrations (IC_50_) of 65.52, 52.91, and 62.54 µM for H460, A549, and H292 cell lines, respectively. However, the IC_50_ values of CTP-(4-OH)-PU were higher than 100 µM for all NSCLC cell lines (Figure 2A). These results indicated that CTPPU was more cytotoxic than CTP-(4-OH)-PU. To determine whether *N,Nʹ*-substituted phenylurea derivatives show selective cytotoxicity between normal and cancer cells. The IC_50_ value of CTPPU and CTP-(4-OH)-PU for normal cells (BEAS-2B and EA.hy926) was higher than 100 µM, suggesting that these compounds had no significant effect on cell viability (Figure 2A). However, only CTPPU showed specific cytotoxicity towards cancer cells rather than normal cells (Figure 2B).

Subsequently, we investigated the long-term efficacy of *N,Nʹ*-substituted phenylurea derivatives against the three NSCLC cell lines through assessment via colony formation assay. As shown in the results, CTPPU-treated H460, A549, and H292 cells lost the capacity to proliferate in a dose-dependent manner (Figure 2C). As shown in Figure 2D, CTPPU treatment (25–50 µM) significantly reduced the plating efficiency and size of colonies for all NSCLC cell lines compared with the control group. By contrast, CTP-(4-OH)-PU had no effect on the plating efficiency or size of colonies for all NSCLC cell lines (Figure 2C,D). Notably, the colony-forming ability of NSCLC cells was markedly reduced by CTPPU at a non-cytotoxic dose of the drug. Thus, CTPPU had a greater potential to suppress NSCLC cell proliferation than CTP-(4-OH)-PU.

### 2.3. N,Nʹ-Substituted Phenylurea Derivatives Inhibit Cell Growth Rather Than Cause Cell Death through Apoptosis of NSCLC Cells

We further investigated the possibility of apoptosis as a mechanism of the cytotoxicity of *N,Nʹ*-substituted phenylurea derivatives. Co-staining with Hoechst 33342 and propidium iodide (PI) was performed on three NSCLC cell lines (H460, A549, and H292). Hoechst 33342 staining was used to evaluate the nuclear morphology of apoptotic cells, which were characterized by condensation and fragmentation of DNA. PI fluorescent dye was utilized to detect necrotic or late apoptotic cells, which were characterized by the loss of integrity of the plasma and nuclear membranes. After NSCLC cell lines were treated with CTPPU or CTP-(4-OH)-PU at various concentrations (0, 25, 50, and 100 µM) and Hoechst 33342 and PI staining, the percentage of apoptotic cells was calculated based on the stained image (Figure 3A). The treatment with the highest concentration of 100 µM CTPPU or CTP-(4-OH)-PU for 24 h can cause the morphological apoptosis of approximately 4–15% cells, and no necrotic cells were detected at the tested CTPPU or CTP-(4-OH)-PU concentrations (Figure 3B; Appendix A).

We confirmed the effect of CTPPU on apoptosis by flow cytometric analysis using Annexin V FITC/PI double staining. NSCLC cell lines (H460, A549, and H292) were treated with various concentrations (0, 25, 50, and 100 µM) of CTPPU for 24 h, stained with Annexin V FITC/PI, and subjected to a flow cytometer. As shown in Figure 4A,B, CTPPU significantly induced early apoptosis of the A549 cells at 25 µM, whereas H460 and H292 exhibited apoptosis at the highest concentration (100 µM). The percentage of apoptotic cells in H460, A549, and H292 cells treated with 100 µM CTPPU was 8.95%, 14.53%, and 4.92%, respectively (Figure 4B; Appendix A). This evidence was consistent with the Hoechst 33342/PI staining results.

Notably, the cell viability of CTPPU-treated cells was inconsistent compared with the percentage of apoptotic cells observed by the MTT assay. The viability of NSCLC cells after the treatment with 100 µM CTPPU was reduced by an average of 50–60%, as shown by the MTT assay (Figure 2A; Appendix A), whereas apoptotic cells accounted for 5–15% (Appendix A). Thus, the percentage of cell viability in NSCLC-treated cells compared with the control was reduced due to the suppression of cell proliferation rather than cell death by apoptosis. These results indicated that CTPPU showed anticancer activity by the suppression of cell growth and the inhibition of cellular proliferation in NSCLC cells.

### 2.4. CTPPU Inhibits Cell Proliferation through Akt/glycogen Synthase Kinase (GSK)-3β/c-Myc Suppression Compared with CTP-(4-OH)-PU in NSCLC Cells

To explore the underlying mechanism responsible for the effect of *N,Nʹ*-substituted phenylurea derivatives, we examined the expression of selected proteins involved in cell proliferation and survival by Western blotting. The Akt signaling pathway regulates a wide range of target proteins that control cell proliferation, survival, growth, and other processes [29]. We then evaluated the effects of *N,Nʹ*-substituted phenylurea derivatives on the reduction of Akt protein levels by Western blot analysis of two NSCLC cell lines (H460 and H292; Figure 5A; Appendix A). Western blot analyses showed that the activation of Akt (p-Akt, phosphorylated at Ser473) and total Akt in H460 and H292 cells decreased in response to CTPPU treatments, especially at a concentration of 25 µM. However, this effect was not observed in CTP-(4-OH)-PU treatment of both NSCLC cell lines (Figure 5A,B). Thus, CTPPU did not cause apoptosis, but it inhibited the growth and cell proliferation of NSCLC cell lines via suppression of the Akt pathway.

To confirm that CTPPU inhibited NSCLC cell proliferation by targeting the Akt signaling pathway, we next aimed to investigate the possible upstream and downstream targets of CTPPU linked with Akt suppression. The Akt signaling pathway is a critical regulator of GSK-3β activation, and it phosphorylates GSK-3β at Ser9 to inactivate GSK-3β [22]. GSK-3β has important roles as both a tumor suppressor (e.g., inhibition of β-catenin, c-Myc, and Mcl-1) and promoter (e.g., inhibition of the cell cycle inhibitor p27^Kip1^) [30]. Here, we investigated whether suppressing Akt by CTPPU led to reduced phosphorylation of GSK-3β and its downstream effector c-Myc. H460 cells were treated with CTPPU or CTP-(4-OH)-PU at various concentrations (0–25 µM) for 24 h, and the expression levels of p-Akt, total Akt, phosphorylated-GSK-3β (Ser9), GSK-3β, β-catenin, and c-Myc were determined by Western blot analysis (Figure 5C; Appendix A). Figure 5D shows the significant decrease in the levels of p-Akt/total Akt, p-GSK-3β/GSK-3β, β-catenin, and c-Myc in H460 cells at 24 h after 25 µM CTPPU treatment. However, no significant decreases in their protein expressions were observed after CTP-(4-OH)-PU treatment. Furthermore, we investigated the possible upstream target regulated by Akt in response to the CTPPU treatments, such as the RTK Met in H460 cells. However, no alteration in the protein expression level of Met was observed after CTPPU treatment in H460 cells (Figure 5C,D).

Comparison of *N,Nʹ*-substituted phenylurea derivatives showed their slightly modified structures. CTPPU contains an *Nʹ*-phenyl substituent, whereas CT-(4-OH)-PU has an *Nʹ*-(4-hydroxyphenyl) substituent. As shown in previous results, the most potent cytotoxicity and antiproliferative effect against NSCLC cells were induced by CTPPU (Figure 2). CTPPU inhibits cell proliferation through the Akt/GSK-3β/c-Myc signaling pathway by decreasing the expression levels of Akt and its downstream effectors, including p-GSK-3β, β-catenin, and c-Myc, whereas CTP-(4-OH)-PU had no effect on such cell growth regulatory signals (Figure 5). These results indicate that different substituents have an effect on the biological antiproliferative activity and that the *Nʹ*-(4-hydroxyphenyl) substituent decreases anticancer activity. Thus, for further action mechanism study, CTPPU was employed as a representative.

### 2.5. CTPPU Inhibits Cell Proliferation by Inducing G1/S Cell Cycle Arrest in NSCLC Cells

Having revealed the proteins involved in CTPPU suppression of NSCLC cell proliferation, we next aimed to investigate the possible signaling pathway from all detected proteins using bioinformatics tools. The Kyoto Encyclopedia of Genes and Genomes (KEGG) pathway (https://www.genome.jp/kegg/mapper.html, accessed on 10 April 2022) was utilized to construct the possible signaling pathway of CTPPU treatment. Meanwhile, CTPPU treatment inhibited cell proliferation by decreasing p-Akt, p-GSK-3β, and c-Myc expression levels (Figure 5C,D). The KEGG pathways related to Akt-regulated cell proliferation, namely, the “PI3K-Akt signaling pathway,” were identified. The KEGG map suggested that the Akt downstream effector c-Myc is involved with cell cycle progression (Figure 6A). As cell proliferation is correlated with the regulation of cell cycle progression [31], and our previous data demonstrated that c-Myc expression was reduced in response to CTPPU treatment, we examined whether the effect of CTPPU led to an alteration in the NSCLC cell cycle. The cell cycle distributions of H460 and H292 cells were investigated with different concentrations of CTPPU (0–50 µM) for 24 h. As shown in Figure 6B, CTPPU at 25 and 50 µM significantly increased the percentage of cell number in the G0/G1 and decreased the percentage of cell number in the S and G2/M phase, indicating that CTPPU inhibited cell cycle progression. Cell cycle analysis revealed a major accumulation of G0/G1 cell populations in H460 (from 44% to 54%) and H292 (from 38% to 46%) in a dose-dependent manner at 24 h after CTPPU treatment (Figure 6B,C; Appendix A). As senescent cells can enter a non-proliferative state and undergo cell cycle arrest, we tested whether CTPPU induced senescence in NSCLC cells. The senescent state of CTPPU-treated cells was confirmed by staining for senescence-associated β-galactosidase (SA-β-gal), the widely used assay for cellular senescence [32]. After 48 h of CTPPU treatment, the percentage of SA-β-gal-positive cells of H460 was increased in a dose-dependent manner (0, 25, and 50 µM; Appendix A).

The results suggest that the Akt suppressed by CTPPU inhibits the proliferation of NSCLC cells through the downregulation of c-Myc, which results in G1/S cell cycle arrest.

### 2.6. Molecular Docking Simulation Reveals the CTPPU Interaction with the Akt and Mammalian Target of Rapamycin Complex 2 (mTORC2) but Not PI3K Protein

Having revealed that CTPPU induced cell cycle arrest and inhibited NSCLC cell proliferation via the Akt signaling pathway, we next aimed to identify possible targets for CTPPU. Molecular docking was used to investigate the possible binding interaction of CTPPU with Akt and its upstream regulators. First, we evaluated the feasibility of direct interaction between CTPPU and Akt-1. We performed a molecular docking simulation of CTPPU with Akt-1 (PDB ID: 3O96). Inhibitor VIII, an Akt-1 allosteric inhibitor, was used as a reference compound. Figure 7A displays the binding affinities of CTPPU and inhibitor VIII. Figure 7B,D shows that inhibitor VIII and CTPPU have the same binding site at the allosteric pocket of Akt-1. CTPPU formed 13 hydrophobic interactions with Gln79, Trp80, Thr82, Ser205, Leu210, Tyr263, Leu264, Lys268, Val270, Tyr272, Asp274, Asp292, and Gly294 with a binding affinity of −9.2 kcal/mol (Figure 7A). Similar to other Akt-1 allosteric inhibitors, pharmacophore modeling reported that Trp80 has a pivotal role in binding Akt-1 to its allosteric manner [33]. The analyses suggested that CTPPU can interact with Akt-1. Furthermore, the effect of CTPPU binding on Akt was validated by immunofluorescence analysis. We investigated whether the effect of CTPPU on p-Akt and total Akt protein levels was similar to that of the specific inhibitor of the PI3K/Akt pathway, wortmannin, which is an irreversible inhibitor of PI3K. H460 cells were treated with CTPPU or wortmannin at the same concentration of 25 µM for 24 h. Then, the p-Akt and total Akt protein levels were determined by immunofluorescence assay (Figure 7E). The immunofluorescence analysis revealed that CTPPU decreased p-Akt and total Akt protein levels, whereas wortmannin decreased only the p-Akt levels and caused no alteration in the total Akt level (Figure 7E,F). This result suggests that CTPPU interacts with Akt and may be involved with Akt stability.

The mammalian target of rapamycin complex 2 (mTORC2) plays an important role in the regulation of the Akt protein life cycle; it initially stabilizes Akt protein folding through turn motif phosphorylation and then promotes Akt protein degradation through hydrophobic motif phosphorylation [34]. To investigate whether mTORC2 can interact with CTPPU, we performed a molecular docking simulation of CTPPU with mTORC2 (PDB ID: 6ZWM). To investigate the novel mTORC2-specific inhibitor, we docked CTPPU into the three small-molecule binding sites of mTORC2 (Figure 8A,C). Figure 8D provides the docking results of CTPPU with a catalytic site, an I-site, and an A-site of mTORC2. For the catalytic site of mTORC2 (Figure 8A), CTPPU formed one hydrogen bond with Lys2187 and 11 hydrophobic interactions with Ser2165, Gln2167, Pro2169, Leu2185, Tyr2225, Ile2237, Gly2238, Trp2239, Val2240, Met2345, and Ile2356 at the catalytic site with a binding affinity of −7.6 kcal/mol. For the A-site (Figure 8B), CTPPU formed one hydrogen bond with Arg576 and seven hydrophobic interactions with Lys541, Asn543, Trp546, Arg572, Arg575, Tyr579, and Phe580 at the A-site of Rictor with a binding affinity of −7.1 kcal/mol. Moreover, binding to the A-site of Rictor may disrupt the mTORC2 activity. For the I-site (Figure 8C), CTPPU formed 12 hydrophobic interactions with Leu1654, Ala1657, Ser1658, Ala1669, Leu1673, Gln1695, Tyr1698, Lys1702, Asn1703, Lys1706, and Arg1749 with a binding affinity of −6.7 kcal/mol. This finding reasonably proves that CTPPU can interact with mTORC2.

PI3K is an upstream regulator of Akt [35]. To investigate whether upstream activators of Akt or PI3Ks can interact with CTPPU, we performed a molecular docking simulation of CTPPU with four isoforms of PI3K, including PI3K-α (PDB ID: 4JPS), PI3K-β (PDB ID: 4BFR), PI3K-δ (PDB ID: 6OCU), and PI3K-γ (PDB ID: 6XRM; Figure 9A–H). Figure 9I lists the binding affinities of CTPPU against the four isoforms of PI3K. All reference compounds showed the highest binding affinities and formed the key hydrogen bond with Val residue at the ATP binding site of PI3Ks (Figure 9A,C,E,G). By contrast, CTPPU showed the lowest binding affinities, and it did not form a hydrogen bond with Val residue (Figure 9B,D,F,H). Our results revealed that CTPPU did not show considerable binding to PI3Ks. We also confirmed by Western blot analysis of H460 cells that CTPPU had no effect on the active PI3K (p-PI3K) compared with the PI3K inhibitor wortmannin. H460 cells were treated with CTPPU, wortmannin, or the combination at the same concentration (25 µM) for 24 h, and the p-PI3K, total PI3K, p-Akt, and total Akt protein levels were determined by Western blot analysis. The result showed a decreased p-PI3K protein level in H460 cells at 24 h after wortmannin treatment, whereas CTPPU treatment had no effect (Figure 9J; Appendix A). The p-PI3K protein level in H460 cells treated with the combination of CTPPU and wortmannin was comparable to that of cells treated with wortmannin alone. However, p-Akt protein levels were dramatically reduced compared to those treated with either wortmannin or CTPPU alone (Figure 9J).

Altogether, these results indicated that CTPPU might inhibit Akt activity and stability via blocking Akt at the allosteric pocket and interacting with mTORC2 to impair its function, respectively, rather than by inhibiting its upstream regulator PI3K.

## 3. Discussion

Advancements in molecular biology and clinical evidence regarding lung cancer have shown a shift in the treatment paradigm toward targeted therapy. Several studies demonstrated the effect of PI3K/AKT and their downstream cascades on the pathogenesis, progression, and treatment failure of the disease [9,11,17]. Akt-inhibiting agents are currently being tested in clinical trials for lung cancer management with promising results [9,17,36]. In our current study, SAR studies comparing the two *N,Nʹ* phenylurea derivatives revealed that CTPPU with *Nʹ*-phenyl substituent is more potent against NSCLC cells than CT-(4-OH)-PU with *Nʹ*-(4-hydroxyphenyl) substituent. The cytotoxic and anti-proliferative effects of *N,Nʹ* phenylurea derivatives were assessed by MTT and colony formation assays. We observed that CTPPU was more effective than CT-(4-OH)-PU at reducing cell viability and colony formation in NSCLC cells. Therefore, CTPPU was used as a representative to better understand the anticancer activity and reveal the possible mechanism underlying the inhibition of the Akt signaling pathway in NSCLC cells.

Akt regulates c-Myc via a GSK-3β-dependent mechanism. Akt inactivates GSK-3β through phosphorylation of Ser9, with a subsequent decrease in c-Myc proteasomal degradation through c-Myc phosphorylation on Thr58 [30,37]. The inactivated GSK-3β stabilizes β-catenin and translocate it into the nucleus, where β-catenin in the nucleus acts as the transcription factor promoting the expression of c-Myc [30]. Thus, GSK-3β may inhibit c-Myc expression through c-Myc transcription and protein stability [37]. In addition, c-Myc stimulates cell cycle progression and cellular proliferation through the regulation of genes related to cell cycle control. c-Myc induces positive cell cycle regulators, including cyclin-dependent kinases, cyclins, and E2F transcription factors [38]. Thus, Akt inhibition resulted in the activation of GSK-3β, which decreased β-catenin expression and c-Myc activities [39]. This evidence agrees with our result showing that CTPPU significantly decreased the expression levels of Akt and its downstream effectors, including p-GSK-3β, β-catenin, and c-Myc, which resulted in G1/S cell cycle arrest and inhibited the proliferation of NSCLC cells.

We also considered the upstream regulators of Akt, such as PI3K, and utilized in silico analysis to predict the possible interaction of the compound at the active sites of the PI3K molecule. Figure 9A–I indicate that the active compound CTPPU did not form the key hydrogen bond with Val residue, whereas reference compounds formed hydrogen bonds with Val residue at the ATP binding site of PI3Ks. In addition, compared with the reference compounds, CTPPU showed the lowest binding affinities for all four isoforms of PI3Ks (PI3Kα, PI3Kβ, PI3Kδ, and PI3Kγ). The molecular docking result was validated by Western blot analysis of the active form of PI3K (phosphorylated PI3K at Tyr485 and Tyr199), which revealed that wortmannin inhibited p-PI3K more efficiently than CTPPU.

Regulation of Akt also depends on the mTORC2 function. mTORC2 mainly promotes cell proliferation and survival through phosphorylation of the AGC kinase family members, including Akt, protein kinase C family members, and serum- and glucocorticoid-induced kinase 1 in their hydrophobic motif (HM) and turn motif (TM) [40]. mTORC2 phosphorylates Akt on Ser473 of the HM site for full Akt activation and Thr450 of the TM site for Akt stability [41]. mTORC2 associates with ribosomes to phosphorylate nascent Akt on its Thr450 during translation [42]. This phosphorylation event prevents premature Akt ubiquitination and increases the stability of nascent Akt polypeptide [42]. Figure 8 indicates the possible interaction of CTPPU with the mTORC2 complex at the catalytic site, I-site, and A-site of mTORC2 with higher binding affinities than the reference compounds, and these bindings may disrupt the mTORC2 activity. Moreover, the compound possibly has an additional effect rather than a direct interaction with the Akt protein.

Akt plays a critical role in cell proliferation and cancer cell growth [9]. In lung cancer cells, KRAS is highly activated, and activated KRAS activates PI3K. The PI3K then promotes Akt phosphorylation on T308 [17]. Other kinases, such as mTORC2 and phosphoinositide-dependent protein kinase-2, phosphorylate Akt at Ser473 [43]. The phosphorylation on Ser473 is important for the full activation of Akt and maintains the phosphorylation on Thr308 [44]. As Akt is a cellular regulator for several functions, such as proliferation, survival, metabolism, and motility [45,46], the agents targeting Akt function and cellular abundance are valuable for cancer therapy. In addition, the inhibition of Akt exerts anticancer effects on lung cancer cells via several modes of cancer cell death. A polyphenol compound, PE5, efficiently suppresses Akt/mTOR, and Bcl-2 induces autophagic cell death in NSCLC [47]. Norcycloartocarpin targeting Akt suppresses cancer cell motility by suppressing the epithelial-mesenchymal transition [48]. Recently, the inhibitor of Akt was shown to have a potential for cancer stem cell inhibition [49]. Although the in vitro experiment is an important step for the identification of a new lead compound that can provide in detail information about the molecular mechanism of action to specific cellular proteins, in the current study, further confirmation on in vivo evaluating model is required for the translational potential of this compound.

In conclusion, this study reported novel information about CTPPU in the suppression of Akt in human NSCLC cells, at least in part by binding with Akt at the allosteric pocket. The Akt inhibition caused by the compound resulted in suppressed cell proliferation. The in silico analysis suggested that CTPPU can interact with Akt-1 and mTORC2 but not PI3Ks. These data can support the potential use of CTPPU in lung cancer therapy.

## 4. Materials and Methods

### 4.1. Synthesis of N,Nʹ-Substituted Phenylurea Derivatives

Unsymmetrically *N,Nʹ*-substituted phenylurea derivatives were prepared via an in situ formed isocyanate of 3-(4-chloro-3-(trifluoromethyl) 1,4,2-dioxazol-5-one **(1)** under the mild condition reported by Chamni and co-workers [28]. Compound **1** was synthesized based on the green protocol reported by Chang and co-workers [50]. Regarding the general protocol for the synthesis of *N,Nʹ*-substituted phenylurea derivatives, dioxazolone **1** (1.5 g, 5.65 mmol) was weighed into an oven-dried reaction vessel equipped with a magnetic stirrer and dissolved in methanol (15 mL). Sodium acetate (Fisher Scientific Company, Fair Lawn, NJ, USA) was added to the reaction (0.47 g, 5.65 mmol) and stirred at room temperature (RT) for 5 min. Then, the corresponding substituted aniline reagent (Tokyo Chemical Industry, Tokyo, Japan) (5.65 mmol) was added. The reaction was heated at 60 °C for 2 h. After completion, the reaction mixtures were concentrated to obtain a crude product, which was further purified by silica gel flash chromatography using a mixture of petroleum ether and ethyl acetate as a mobile phase (Merck, Darmstadt, Germany). The chemical structure of a purified product was identified based on spectroscopic analysis. The obtaining chemical data of selected *N,Nʹ*-substituted phenylurea derivatives matched with the previous report [28].

*N*-[4-chloro-3-(trifluoromethyl)phenyl]-*Nʹ*-phenylurea (CTPPU) was obtained as a white solid in 83% yield (1.48 g) using dioxazolone **1** and aniline (0.53 g, 5.65 mmol, **2a**). ^1^H NMR (500 MHz, DMSO-*d*_6_): δ 9.15 (1H, s), 8.83 (1H, s), 8.10 (1H, d, *J* = 2.5 Hz), 7.63 (2H, m), 7.46 (2H, d, *J* = 8.5 Hz), 7.29 (2H, t, *J* = 8.0 Hz), 7.00 (1H, t, *J* = 7.5 Hz); ^13^C NMR (125 MHz, DMSO-*d*_6_): δ 152.2, 139.2, 139.0, 131.8, 128.6 (2C), 126.7 & 126.4 (d, *J* = 30.0 Hz), 123.7 & 121.6 (d, *J* = 271.3 Hz), 122.9, 122.1, 118.5 (2C), 116.6 & 116.5 (d, *J* = 17.5 Hz), 116.6 & 116.5 (d, *J* = 5.0 Hz).

*N*-[4-chloro-3-(trifluoromethyl)phenyl]-*Nʹ*-(4-hydroxyphenyl)urea (CTP-(4-OH)-PU) was afforded as a white powder in 80% yield (1.49 g) using dioxazolone **1** and 4-aminophenol (0.62 g, 5.65 mmol, **2b**). ^1^H NMR (500 MHz, DMSO-d_6_): δ 9.13 (1H, s), 9.03 (1H, s), 8.49 (1H, s), 8.09 (1H, d, *J* = 2.5 Hz), 7.60 (2H, m), 7.22 (2H, d, *J* = 9.0 Hz), 6.69 (2H, d, *J* = 9.0 Hz); ^13^C NMR (125 MHz, DMSO-d_6_): δ 153.4, 153.1, 140.1, 132.4, 130.9, 127.2 & 127.0 (d, J = 31.3 Hz), 123.3, 121.5 (2C), 117.0, 117.0, 117.0 & 116.9 (d, *J* = 5.0 Hz), 115.7 (2C).

### 4.2. Reagents and Antibodies

Roswell Park Memorial Institute (RPMI) 1640 medium, Dulbecco’s Modified Eagle’s Medium (DMEM), fetal bovine serum (FBS), Antibiotic-Antimycotic (#15240096), GlutaMAX™ l-alanyl-l-glutamine dipeptide (#35050061), phosphate-buffered saline (PBS; pH 7.4), and 0.25% trypsin-EDTA were obtained from Gibco (Gaithersburg, MA, USA). 3-(4,5-dimethylthiazol-2-yl)-2,5-diphenyltetrazolium bromide (MTT), dimethyl sulfoxide (DMSO), Triton X-100, Ribonuclease A (RNase A), crystal violet, paraformaldehyde, wortmannin, Hoechst 33342 and propidium iodide (PI) were purchased from Sigma-Aldrich Chemical, Co. (St. Louis, MO, USA). A radioimmunoprecipitation assay (RIPA) lysis buffer and Immobilon Western Chemiluminescent HRP Substrate were purchased from Merck (Darmstadt, Germany), and the protease inhibitor cocktail was purchased from Roche Molecular Biochemicals (Indianapolis, IN, USA). A bicinchoninic acid (BCA) protein assay kit was purchased from Thermo scientific (Rockford, IL, USA). The primary antibodies against p-PI3K (Tyr485/Tyr199; #4228), total PI3K (#4292), p-Akt (Ser473; #4060), total Akt (#9272), Met (#8198), p-GSK-3β (Ser9; #9332), total GSK-3β (#9832), β-catenin (#8480), c-Myc (#5605), β-actin (#4970), and the secondary antibody anti-rabbit IgG (#7074) or anti-mouse IgG (#7076) were acquired from Cell Signaling Technology (Danvers, MA, USA).

### 4.3. Cell Culture

Human non-small-cell lung cancer (NSCLC)-derived (H460, H292, and A549 cells), human lung epithelial (BEAS-2B) cells and human vascular endothelial cells (EA.hy926) were purchased from the American Type Culture Collection (Manassas, VA, USA). H460 and H292 cells were cultured in an RPMI medium. A549, BEAS-2B and EA.hy926 cells were cultured in DMEM medium. The medium was supplemented with 10% FBS, 1% GlutaMAX™, and 1% Antibiotic-Antimycotic. Cells with 70–80% confluence were trypsinized with 0.25% trypsin-EDTA and subcultured in the same media. Some aliquots of cells were transferred to a freezing medium containing 10% (*v*/*v*) DMSO and 50% (*v*/*v*) FBS and stored at −80 °C for later use.

### 4.4. Preparation of Compounds Solution

A stock concentration of 50 mM CTPPU and CTP-(4-OH)-PU were dissolved in dimethyl sulfoxide (DMSO; Sigma-Aldrich Chemical) and stored in aliquots at −20 °C until use. CTPPU and CTP-(4-OH)-PU with designated final concentrations (0–100 µM) was diluted with cell culture media for subsequence experiments (with a maximal DMSO concentration less than 0.5% DMSO). Wortmannin, a PI3K inhibitor, was dissolved in DMSO at a concentration of 25 mM and stored at −20 °C.

### 4.5. Cytotoxicity Assay

Cytotoxicity assay was performed to evaluate cell growth inhibition of the two compounds of *N,Nʹ*-substituted phenylurea derivatives (CTPPU and CTP-(4-OH)-PU) on three NSCLC cells (H460, A549, and H292) and normal cells (BEAS-2B and EA.hy926). The cytotoxicity of test compounds in NSCLC cells was assessed by using the MTT assays. Briefly, the cells were seeded in 96-well plates at 7.5–10 × 10^3^ cells/well and incubated for 24 h. Next, the cells were treated with various concentrations (0–100 µM) of CTPPU and CTP-(4-OH)-PU for 24 h. The cells were then incubated with 0.5 mg/mL of MTT solution at 37 °C in a dark place for 3 h. The MTT solution was replaced with DMSO (150 µL/well) to dissolve the purple formazan crystal. The intensity of formazan color was measured by optical density (OD) at 570 nm using a microplate reader (PerkinElmer Inc., Waltham, MA, USA). The percentage of cell viability was calculated by following this equation: (OD of treated cell/OD of non-treated control cells) × 100. IC_50_ values were calculated using regression analysis from dose-response curves (GraphPad Prism 7 software, San Diego, CA, USA).

### 4.6. Colony Formation Assay

To check for the colony formation capacity. The cells were seeded in 6-well plates at 300 cells per well. Following overnight attachment, cells were treated with various concentrations (0–50 μM) of CTPPU or CTP-(4-OH)-PU for 24 h. After treatment, the colonies were allowed to form for 7 days, and the medium was changed every two days. For colony staining, colonies were washed once with PBS, then fixed by adding fixative (methanol: acetic acid (3:1, *v*/*v*)) for 5 min and stained with crystal violet (0.05% (*w*/*v*)) in 4% paraformaldehyde for 30 min. The excess crystal violet was washed with distilled water several times and let air dry at RT. The colonies were photographed with a digital camera, and the acquired images were analyzed using the ImageJ software (National Institutes of Health (NIH), Bethesda, MD, USA).

### 4.7. Hoechst 33342 and PI Staining Assay

Apoptosis was analyzed by co-staining with Hoechst 33342 and PI. Nuclear morphology was assessed using the cell membrane penetration DNA dye Hoechst 33342. The NSCLC cell lines (H460, A549, and H292) were seeded in 96-well plates at 1 × 10^4^ cells/well and treated with various concentrations (0–100 µM) of CTPPU and CTP-(4-OH)-PU for 24 h. The cells were stained with 10 µg/mL Hoechst 33342 and 1 µg/mL PI for 30 min at 37 °C. Then, the cells were visualized under a fluorescent microscope (Nikon ECLIPSE Ts2, Tokyo, Japan), and the percentage of apoptotic cells was calculated by following this equation: (Total number of apoptotic bodies/Total number of cells count) × 100.

### 4.8. Apoptosis Analysis

The apoptotic cells were measured using Annexin V-FITC Apoptosis Kit (ImmunoTools, Friesoythe, Germany) according to the manufacturer’s instructions. Briefly, NSCLC cell lines (H460, H292 and A549) were seeded in a 24-well plate at 5 × 10^4^ cells/well for 24 h. After being treated with various concentrations of CTPPU for 24 h, the cells were collected and washed twice with PBS. The pellet was then re-suspended in 100 μL of binding buffer containing 2.5 μL of AnnexinV-FITC and 1 μL of PI and then incubated at RT for 15 min in the dark. The cells were analyzed by flow cytometer using Guava easyCyte HT (Luminex Corporation, Austin, TX, USA). The results were analyzed using guava InCyte software (Luminex Corporation).

### 4.9. Cell Cycle Analysis

To determine the cell cycle distribution, H460 and H292 cells at 1 × 10^5^ were seeded into 6-well plates and cultured overnight, prior to starvation, in a serum-free medium for 48 h to be synchronized. After serum starvation, cells were treated with various concentrations of CTPPU (0, 25, and 50 µM) for 24 h and harvested. The cells were washed with PBS, then fixed with ice-cold 70% ethanol and incubated overnight at −20 °C. The cells were then washed twice with PBS to remove ethanol and labeled with PI by incubating with PI solution (20 μg/mL PI, 0.1% (*v*/*v*) Triton X-100, and 20 μg/mL RNase A) in the dark for 30 min. The stained cells were analyzed by flow cytometry on a Guava easyCyte HT flow cytometer (Luminex Corporation). The particular phase of the cell cycle with DNA content in sub G1, G0/G1, S, and G2/M was estimated using guava InCyte software (Luminex Corporation).

### 4.10. Senescence-Associated β-Galactosidase (SA-β-Gal) Staining

The positive blue staining of SA-β-gal was used to estimate cellular senescence. The H460 cells (5 × 10^3^ cells/well) were seeded on 96-well plates and incubated overnight. After being treated with various concentrations (0, 25, and 50 µM) of CTPPU for 48 h, cells were stained with the Senescence-Galactosidase Staining Kit (Cell Signaling Technology), following the same method as described previously [51]. Five random fields were captured from each well, and the total number of cells and SA-β-gal positive cells were determined using ImageJ software (NIH, Bethesda, MD, USA).

### 4.11. Protein and Ligand Preparation

The 3D structures of Akt-1 (PDB ID: 3O96) [33], mTORC2 (PDB ID: 6ZWM) [52], PI3K-α (PDB ID: 4JPS) [53], PI3K-β (PDB ID: 4BFR) [54], PI3K-δ (PDB ID: 6OCU) [55], and PI3K-γ (PDB ID: 6XRM) [56] were received from the Research Collaboratory for Structural Bioinformatics Protein Data Bank [57]. All protein structures were prepared for docking using the UCSF ChimeraX [58]. The ligand structure of CTPPU was downloaded from the PubChem Database (PubChem CID: 1278778) [59] and optimized by density functional theory (DFT) with a B3LYP/6–31G (d,p) basis set using the Gaussian 09 program [60]. The PDBQT format was generated by the prepare_receptor and prepare_ligand programs from the AutoDockFR suite [61].

### 4.12. Computational Molecular Docking

Autodock Vina version 1.2.0 [62] was performed to investigate the molecular interaction of ligands and proteins. The exhaustiveness parameter was set as 32, and the grid size was set to 20 × 20 × 20 Å with a spacing of 1 Å. The binding site was set with the center of the ligand from the PDB structure. Furthermore, 3D binding interactions were visualized through the UCSF ChimeraX [58].

### 4.13. Immunofluorescence Assay

H460 cells were seed overnight in 96-well plates at the density of 1 x 10^4^ cells/well. Then, the cells were treated with CTPPU or wortmannin at a concentration of 25 µM and incubated for 24 h. After that, cells were fixed with 4% paraformaldehyde for 15 min, permeabilized by 0.2% (*v*/*v*) Triton X-100 in PBS and blocked with 10% (*v*/*v*) FBS for 20 min at RT. Primary antibody of p-Akt or Total Akt at proportional 1:100 in 4% (*v*/*v*) FBS was applied to the cells before incubation overnight at 4 °C. After incubation time, Alexa Fluor 488 or Alexa Fluor 594 conjugated with goat anti-rabbit IgG secondary antibody at proportional 1:500 in 4% (*v*/*v*) FBS was added and incubated in the dark for 1 h at RT. Cell nuclei were stained with Hoechst 33342 for 15 min at RT and then visualized under a fluorescent microscope with 400× magnification (Nikon ECLIPSE Ts2, Tokyo, Japan). The acquired images were analyzed using the ImageJ software (NIH, Bethesda, MD, USA).

### 4.14. Western Blot Analysis

A total of 3.5 × 10^5^ cells were seeded into 6-well plates and cultured for 24 h. Then, CTPPU or CT-(4-OH)-PU (0–25 µM) was added and cultured for a further 24 h. Cells were washed with ice-cold PBS, lysed in RIPA lysis buffer (Merck) containing protease inhibitor cocktail (Roche Molecular Biochemicals), and centrifuged (12,000 rpm, 20 min, 4 °C). The lysates were collected, and the protein contents were determined by a BCA protein assay kit (Thermo scientific). Equal amounts of total proteins (50 µg) were subjected to 7.5–10% SDS–polyacrylamide gel electrophoresis and transferred to a polyvinylidene difluoride membrane (PVDF; Bio-Rad Laboratories Inc., Hercules, CA, USA). The membranes were blocked in 20 mM Tris-buffered saline-0.05% Tween-20 (TBST) buffer containing 5% (*v*/*v*) skim-milk at RT for 1 h, washed 3 times with TBST for 10 min, and probed with specific primary antibodies at a dilution of 1:1000 at 4 °C overnight. After washing 3 times with TBST for 10 min, the membranes were incubated with horseradish peroxidase (HRP)-conjugated anti-rabbit or anti-mouse IgG for 1 h at a dilution of 1:5000 at RT and washed 3 times with TBST for 10 min. To ensure equal protein loading, β-actin was used as an internal control. Proteins probed with specific primary antibodies were visualized with an enhanced chemiluminescence reagent (Immobilon Western Chemiluminescent HRP Substrate; Merck), and the bands were captured using ImageQuant LAS 4000 (GE Healthcare, Piscataway, NJ, USA); protein bands were quantified using ImageJ Software (NIH, Bethesda, MD, USA).

### 4.15. Statistical Analysis

All experiments were performed at least 3 times, and all results are expressed as the mean ± standard deviation. Statistical analyses were performed using GraphPad Prism 7 (San Diego, CA, USA). The unpaired t-test was used for the statistical analyses between the 2 groups. *p* < 0.05 was considered statistically significant.

## 5. Conclusions

This study suggested that CTPPU suppresses NSCLC cell proliferation by inducing G1/S phase cell cycle arrest by suppressing the Akt/GSK-3β/c-Myc signaling pathway. In addition, docking simulation revealed that the CTPPU molecule could exert allosteric inhibition of the Akt protein and mTORC2 by interrupting its activity to activate and stabilize Akt. As Akt is critical for cancer cell survival and proliferation, our results may be used to demonstrate CTPPU as a potential therapy for lung cancer and Akt-driven cancers.

## Figures and Tables

**Figure 1 ijms-24-01357-f001:**
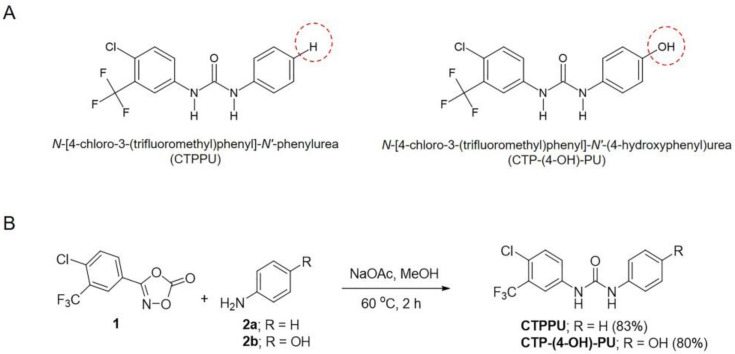
Chemical structure of *N,Nʹ*-substituted phenylurea derivatives and synthesis. (**A**) Structure of CTPPU (left) and CTP-(4-OH)-PU (right). (**B**) Synthetic scheme of *N,Nʹ*-substituted phenylurea derivatives (CTPPU and CTP-(4-OH)-PU) and their yield percentages.

**Figure 2 ijms-24-01357-f002:**
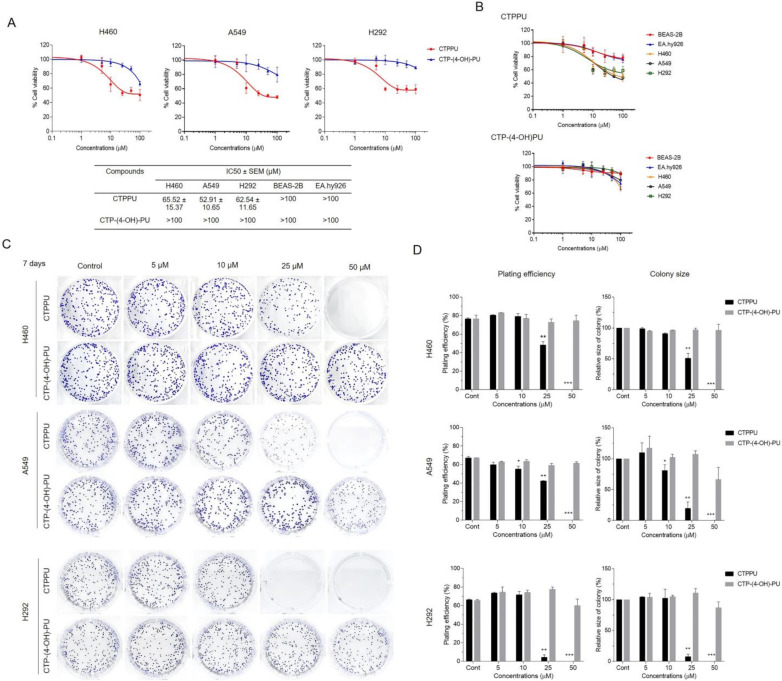
Effects of *N,Nʹ*-substituted phenylurea derivatives on NSCLC cell viability and proliferation. (**A**) Cytotoxic effects of *N,Nʹ*-substituted phenylurea derivatives on NSCLC cell (H460, A549, and H292) and non-malignant cell (BEAS-2B and EA.hy926) line. Cells were treated with various concentrations (0–100 µM) of CTPPU or CTP-(4-OH)-PU for 24 h. Viability was quantitated by MTT assay. (**B**) Compare the cytotoxicity of *N,Nʹ*-substituted phenylurea derivatives on cancer cells vs. normal cells. (**C**) Growth inhibition effects of CTPPU and CTP-(4-OH)-PU on NSCLC cell lines were measured by colony formation assay. (**D**) Bar graphs showed the quantitative results of C; the left part shows the plating efficiency, and the right part shows the colony size after the indicated treatment. Each value is the mean ± SD from triplicate samples. * *p* < 0.05, ** *p* < 0.01, and *** *p* < 0.001 compared with the control.

**Figure 3 ijms-24-01357-f003:**
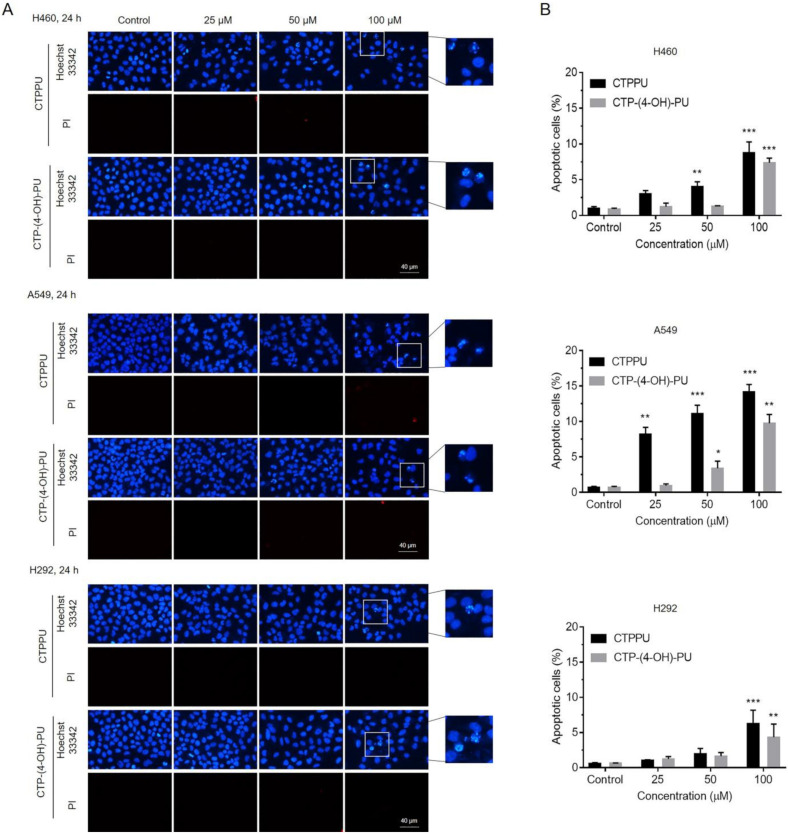
Effects of *N,Nʹ*-substituted phenylurea derivatives treatment on apoptosis-related morphological changes in NSCLC cells. (**A**) Morphological changes visualized by fluorescence microscopy after Hoechst 33342/PI staining. NSCLC cell lines (H460, A549 and H292) were treated with CTPPU or CTP-(4-OH)-PU, stained with Hoechst 33342 and PI at 24 h. All images were captured at 400× magnification. Scale bars, 40 µm. (**B**) The percentages of apoptotic cells were counted in three independent random areas. The results are presented as mean ± SD, and all samples were measured independently in triplicate. Error bars are standard deviations. Significant differences are indicated as * *p* < 0.05, ** *p* < 0.01 and *** *p* < 0.001 compared with the control.

**Figure 4 ijms-24-01357-f004:**
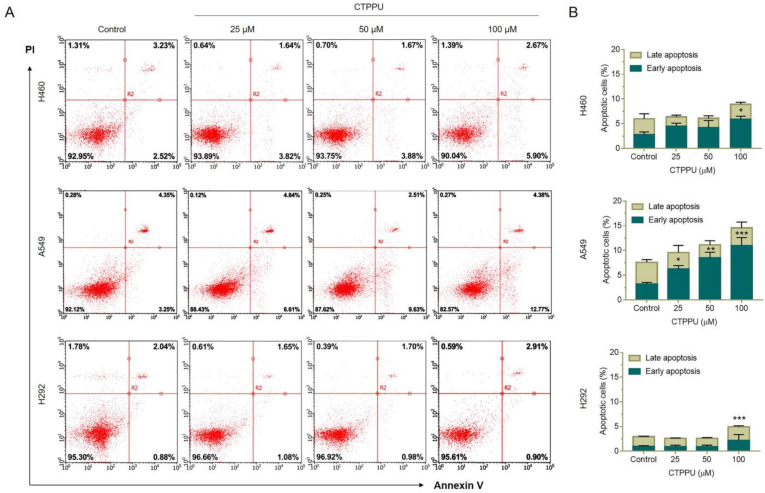
Effect of CTPPU treatment on cell apoptosis in NSCLC cells. (**A**) Apoptotic cells were evaluated using Annexin V FITC/PI double staining and analysis using flow cytometry. Representative dot plots demonstrated the percentage of early apoptotic cells in the lower right quadrant (Annexin+/PI–) and late apoptotic cells in the upper right quadrant (Annexin+/PI+) at various concentrations of CTPPU. (**B**) Bar graphs showed the quantitative results of A. The results are presented as mean ± SD, and all samples were measured independently in triplicate. Error bars are standard deviations. Significant differences are indicated as * *p* < 0.05, ** *p* < 0.01, and *** *p* < 0.001 compared with the control.

**Figure 5 ijms-24-01357-f005:**
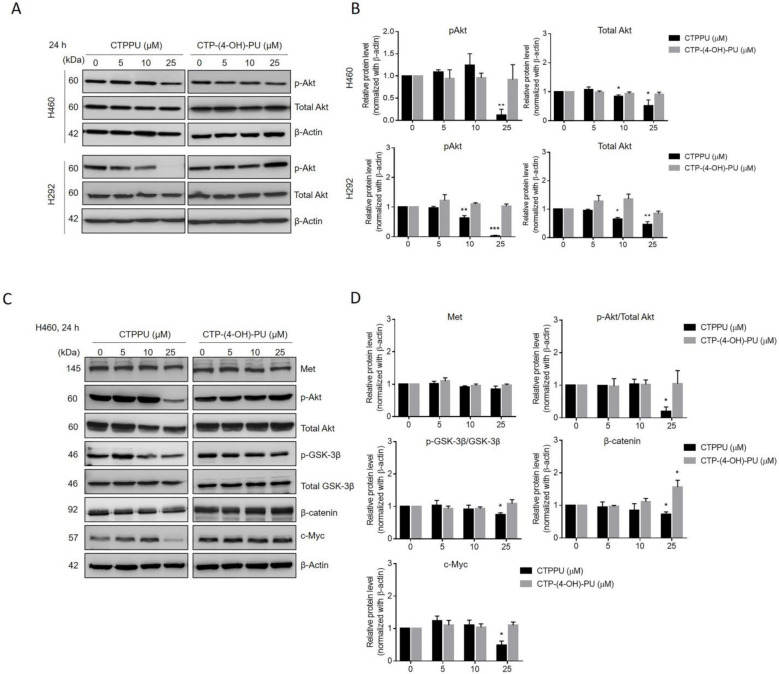
Effect of phenyl urea derivatives (CTPPU and CTP(4-OH)-PU) on Akt/GSK-3β/c-Myc signaling pathway. (**A**) The expression levels of phosphorylated (p) and total Akt in H460 and H292 cells were assessed by Western blot analysis after 24 h of urea derivatives treatment. (**B**) Bar graphs showed the quantitative results of (**A**). (**C**) The expression of Met, p-Akt, total Akt, p-GSK-3β, total GSK-3β, β-catenin, and c-Myc in H460 were determined by Western blotting after being treated with phenyl urea derivatives. β-Actin was used as an internal control. (**D**) Bar graphs showed the quantitative results of C. The data are shown as the mean ± SD of three independent experiments (* *p* < 0.05, ** *p* < 0.01, and *** *p* < 0.001 compared with the control).

**Figure 6 ijms-24-01357-f006:**
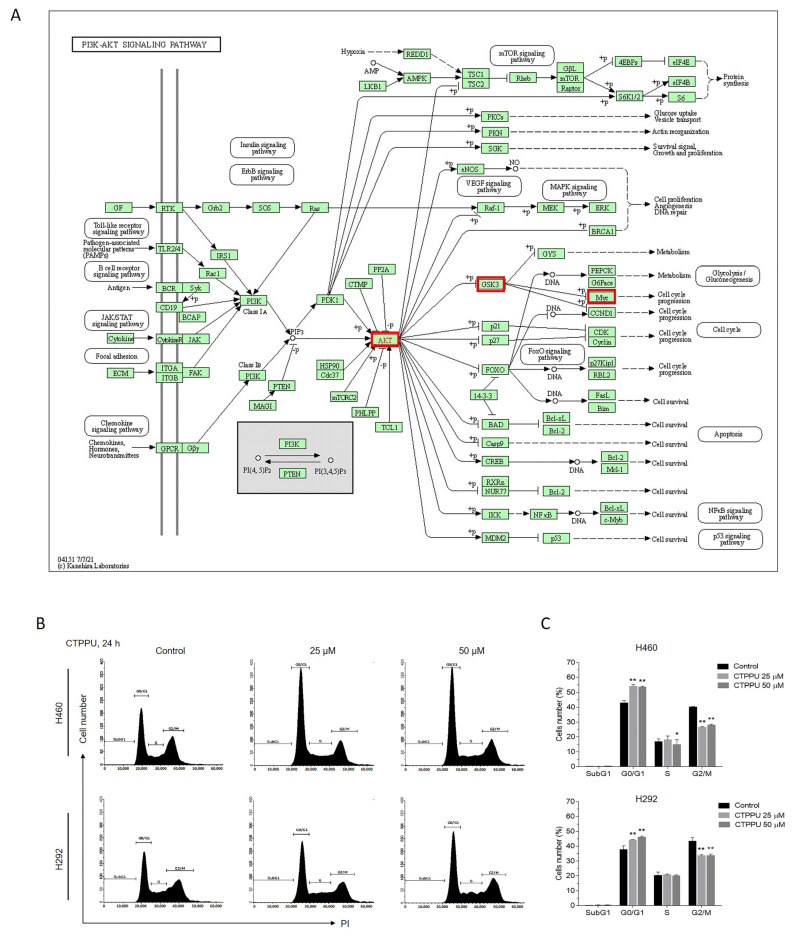
Effect of CTPPU on cell cycle progression of NSCLC cells. (**A**) The KEGG mapper database revealed potential Akt downstream signals involved in cell cycle progression. The red box represents the proteins affected by CTPPU treatment. (**B**) After serum starvation of 48 h, cells were treated with 25 or 50 μM of CTPPU, as indicated, for 24 h. The cell cycle distribution was analyzed through flow cytometry with PI staining. (**C**) The percentage of cells number in each phase of the cell cycle (sub G1, G0/G1, S, and G2/M) is indicated. Each value is the mean (±SD) from triplicate samples. * *p* < 0.05 and ** *p* < 0.01 compared with the control.

**Figure 7 ijms-24-01357-f007:**
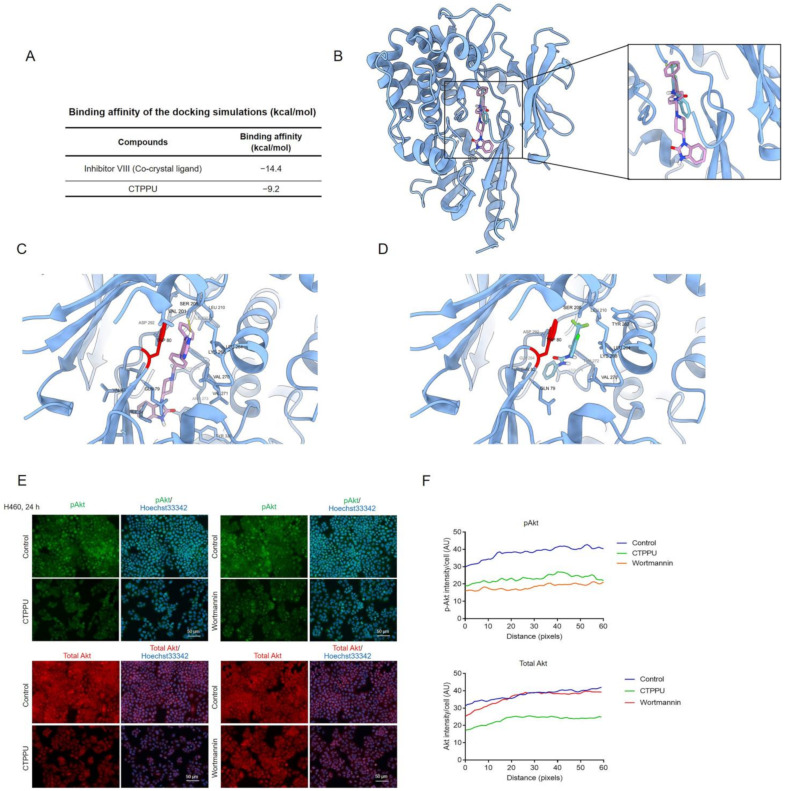
The effect of CTPPU decreased Akt protein expression and a docked model depicting the interaction of CTPPU with the Akt protein. (**A**) The binding affinities of CTPPU against the Akt-1. Docking interaction profile of Akt-1 inhibitors: (**B**) Binding of Inhibitor VIII and CTPPU to the Akt-1 binding site, (**C**) Binding interaction of Inhibitor VIII-Akt-1 complex, (**D**) Binding interaction of CTPPU-Akt-1 complex. (**E**) The expression levels p-Akt and Total Akt were determined by immunofluorescence analysis after CTPPU treatment for 24 h compared to wortmannin. All images were captured at 200× magnification. Scale bars, 50 µm. (**F**) Histogram showed the quantitative results of E. The fluorescence intensity was analyzed using ImageJ software.

**Figure 8 ijms-24-01357-f008:**
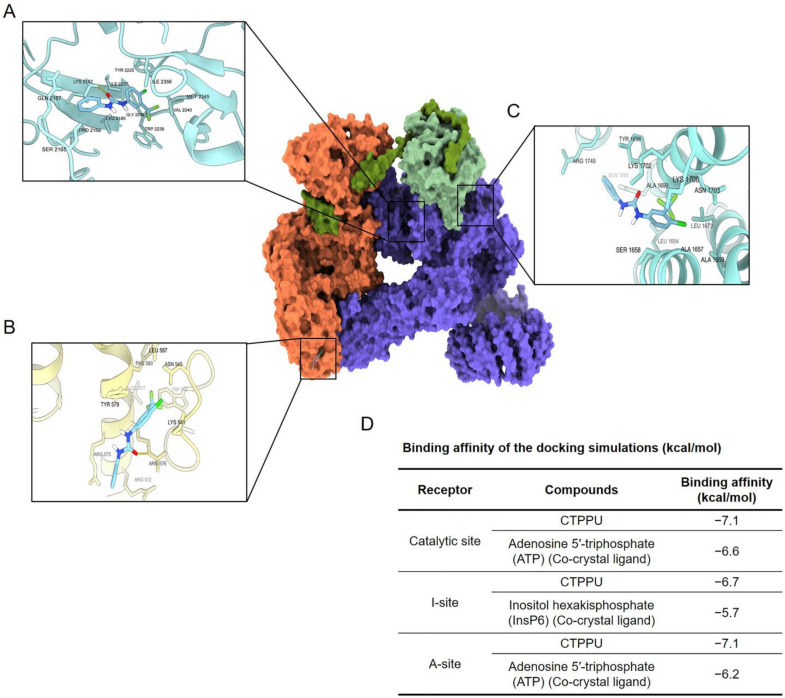
Binding interaction of CTPPU in complex with mTORC2. Docking interaction profile of CTPPU in complex with mTORC2: (**A**) the catalytic site of mTORC2, (**B**) the A-site, (**C**) the I-site. (**D**) The binding affinities of CTPPU against the mTORC2.

**Figure 9 ijms-24-01357-f009:**
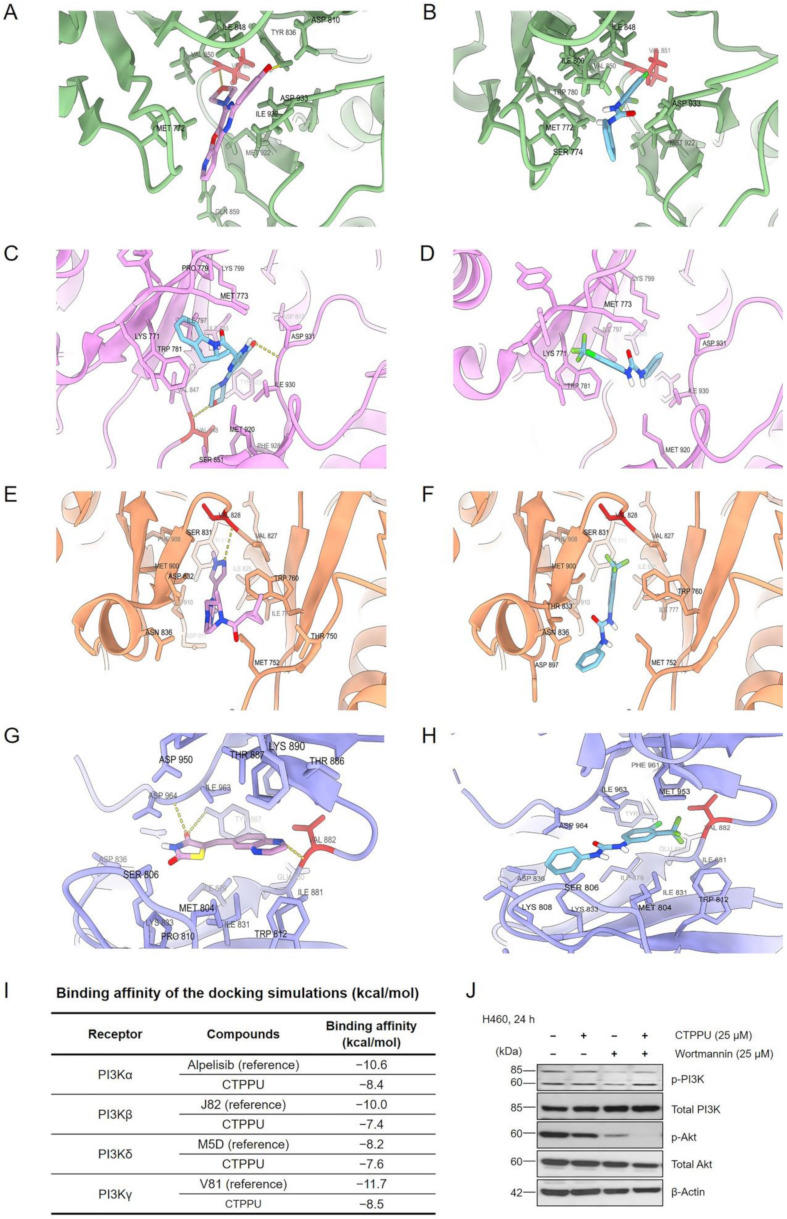
Binding interaction of CTPPU in complex with PI3Ks. Docking interaction profile of PI3Ks inhibitors: (**A**) Alpelisib-PI3Kα complex, (**B**) CTPPU-PI3Kα complex, (**C**) J82-PI3Kβ complex, (**D**) CCTPU-PI3Kβ complex, (**E**) M5D-PI3Kδ complex, (**F**) CTPPU-PI3Kδ complex, (**G**) V81- PI3Kγ complex, and (**H**) CTPPU- PI3Kγ. (**I**) The binding affinities of CTPPU against the 4 isoforms of PI3K (**J**) The expression of p-PI3K, total PI3K, p-Akt, and total Akt in H460 were determined by Western blotting after being treated with CTPPU, wortmannin, or combination. β-Actin was used as an internal control.

## Data Availability

The datasets used and/or analyzed during the current study are available from the corresponding author upon reasonable request.

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
