# Peer review of "N,Nʹ-Diarylurea Derivatives (CTPPU) Inhibited NSCLC Cell Growth and Induced Cell Cycle Arrest through Akt/GSK-3β/c-Myc Signaling Pathway"

_ijms, 2023, doi:10.3390/ijms24021357_

Round 1

Reviewer 1 Report (Previous Reviewer 2)

I previously reviewed the article written by Sunisa Thongsom et al. presenting diarylurea derivatives, NSCLC cells growth and mechanism of their actions. To find out the reason for the rejection of the article, I read other reviews as well as the authors' responses. It is a pity that earlier the authors did not improve their work sufficiently to be accepted for publication.

After carefully reading the manuscript, I confirm that the authors took into account the most important comments of the reviewers and performed appropriate research, such as:

the effect of CTPPU on apoptosis by flow cytometric analysis using Annexin V FITC/PI double staining,

the senescent state of CTPPU-treated cells was confirmed by staining for senescence-associated β-galactosidase (SA-β-gal), the widely used assay for cellular senescence.

The authors also confirmed using Western blot analysis of H460 cells that CTPPU had no effect on the active PI3K (p-PI3K) compared with the PI3K 373 wortmannin inhibitor.

Also the size of the Hoechst coloring images is sufficient

I have no additional comments and I can say that now the manuscript has gained even more scientific value and that the article can be published in its current form. This is a really good work.

Author Response

We appreciate the encouraging comments.

Reviewer 2 Report (New Reviewer)

1.       In Figure-2 . the authors have used two cell lines to access the cytotoxic potential of the compounds to the normal cells. BEAS-2B cells belong to the airway epithelial cells and are relevant to the study, but EA.hy926 cell line is used in cardiovascular system research as evident from Atcc website. But here authors are studying the role of compounds in NSCLS. Please explain the reason behind using this cell line with references.

2.       Line 41, space is missing between ‘two-folds’.

3.       Please state limitations of the study

Author Response

This manuscript is a resubmission of an earlier submission. The following is a list of the peer review reports and author responses from that submission.

Round 1

Reviewer 1 Report

-          Overall, the language of the manuscript is poor. It cannot be published in this current format unless extensively reviewed by a native speaker.

-          The abstract misses to refer to the use of phenylurea derivatives in the background component. The rationale behind using these compounds must be included.

-          The introduction fails to include previous studies on the use of natural compounds to target the PI3K/Akt pathway and relies mainly on the authors’ previous work.

-          The chemistry section of the results is better if moved to the methods section.

-          The MTT and clonogenic survival assays have no value if not coupled with studies in normal lung epithelium (or other types of non-malignant cells) to demonstrate the selective toxicity of CTPPU and CTP-(4-OH)-PU against tumor cells.

-          Graphs in Figure 2C must be re-plotted in a manner that considers plating efficiency.

-          It seems like that CTPPU had a greater ability to inhibit colony forming of H292 cells than its ability to reduce viability at the same concentrations. How can the authors explain that?

-          The magnification of the Hoechst staining images is not mentioned in the figure legend, but they were clearly taken at low magnification. How was the evaluation of nuclear fragmentation done at this low power? Was there a positive control included in this experiment? The use of Hoechst is most definitely not sufficient to rule out apoptosis. More specific markers such as annexin V, TUNEL or measuring the expression of certain components of the apoptosis-regulating machinery is required to support the authors’ conclusion.

-          Lines 165-168: to conclude that the response is primarily growth arrest rather than cell death is premature since other types of cell stress responses were not excluded (cytotoxic autophagy, senescence, ferroptosis, diapause, etc…)

-          Figure 4A: In H460 cells, it seems that p-Akt levels increase at 10 uM CTPPU but then drop dramatically at 25 uM. How do the authors explain these somewhat unexpected patterns of expression?

-          Claiming that the effect of CTPPU is mediated through the Akt pathway without sufficient genetic and pharmacological manipulation of the system is invalid.

The work lacks any in vivo studies to support the cell culture effects of the tested compounds.

Reviewer 2 Report

The article written by Sunisa Thongsom et al. presents diarylurea derivatives, NSCLC cells growth and mechanism of their actions. The article is methodologically very interesting, but I have some important questions which should be addressed by the authors in the revised version of the manuscript.

1. The authors should explain why did they choose CTPPU and CTP-(4-OH)-PU derivatives for further study, while the CTPPU IC50 values for H460, A549, and H292 are 65.5, 52.9, and 62.6 μM, respectively, and those of CTP-(4-OH)-PU were more than 100 μM for all NSCLC cell lines. These values indicate that the tested compounds are inactive against the studied cancer cell lines.

2. Why the toxicity of the derivatives was not tested on a healthy human cell line?

3. The obtained activities are not compared with a known standard. I think Act Inhibitor VIII might be a good standard.

4. The authors found that only the treatment of the highest concentration of 100 μM CTPPU or CTP-(4-OH)-PU for 24 h could cause morphological apoptosis of about 2-4% cells, which confirms their weak activity.

5. Please remove the empty Section 6. Patents.

Round 2

Reviewer 1 Report

The authors have improved the quality of the writing and edited the Introduction section in a manner that better provides rationale for the study. The authors have succeeded in showing that CTPPU is selectively toxic to tumor cells by running viability assays in two non-malignant cell lines.

However, the authors did not successfully address some of the other key issues raised in the first cycle of review:
- Instead of providing higher microscopic magnification of Hoechst staining, they cropped and maximized the original images.
- Excluding the possibility of cells death cannot be made solely on nuclear staining (as its highly non-specific). The authors were invited to include more evidence on more relevant apoptosis/cell death markers such as Annexin V or cleavage of caspase-3 or PARP, but such evidence was not included.
- When asked to consider other mechanisms of cell death, the authors concluded that the observed G1/S growth arrest induced by CTPPU might reflect a state of senescence, but provide no evidence to support it.
- The authors were asked to include pharmacological or genetic inhibition of the PI3K/Akt pathway to confirm the mechanism of action of CTPPU. For that, they employed wortomonin, a known inhibitor of the pathway, but the experimental design strangely omitted a critical condition of combining CTPPU with wortomonin making the data inconclusive.

Reviewer 2 Report

The authors took my comments into account and the manuscript can now be published in its present form.